# PLGA-Encapsulated *Haemonchus contortus* Antigen ES-15 Augments Immune Responses in a Murine Model

**DOI:** 10.3390/vaccines11121794

**Published:** 2023-11-30

**Authors:** Muhammad Waqqas Hasan, Muhammad Ehsan, Qiangqiang Wang, Muhammad Haseeb, Shakeel Ahmed Lakho, Ali Haider, Mingmin Lu, Lixin Xu, Xiaokai Song, Ruofeng Yan, Xiangrui Li

**Affiliations:** 1MOE Joint International Research Laboratory of Animal Health and Food Safety, College of Veterinary Medicine, Nanjing Agricultural University, Nanjing 210095, China; 2015207037@njau.edu.cn (M.W.H.); muhammad.ehsan@iub.edu.pk (M.E.); 20193050436@cau.edu.cn (Q.W.); 2016207041@njau.edu.cn (M.H.); 2017207046@njau.edu.cn (S.A.L.); 2018207074@njau.edu.cn (A.H.); mingmin.lu@njau.edu.cn (M.L.); xulixin@njau.edu.cn (L.X.); songxiaokai@njau.edu.cn (X.S.); 2Department of Parasitology, Faculty of Veterinary and Animal Sciences, The Islamia University of Bahawalpur, Punjab 63100, Pakistan

**Keywords:** *Haemonchus contortus*, rHcES-15, PLGA, nanoparticles, vaccination, mice

## Abstract

*Haemonchus contortus* is a gastrointestinal parasite that adversely impacts small ruminants, resulting in a notable reduction in animal productivity. In the current investigation, we developed a nanovaccine by encapsulating the recombinant protein rHcES-15, sourced from the excretory/secretory products of *H. contortus*, within biodegradable poly (D, L-lactide-co-glycolide) (PLGA) nanoparticles (NPs). The development of this nanovaccine involved the formulation of PLGA NPs using a modified double emulsion solvent evaporation technique. Scanning electron microscopy (SEM)verified the successful encapsulation of rHcES-15 within PLGA NPs, exhibiting a size range of 350–400 nm. The encapsulation efficiency (EE) of the antigen in the nanovaccine was determined to be 72%. A total of forty experimental mice were allocated into five groups, with the nanovaccine administered on day 0 and the mice euthanized at the end of the 14-day trial. The stimulation index (SI) from the mice subjected to the nanovaccine indicated heightened lymphocyte proliferation (*** *p* < 0.001) and a noteworthy increase in anti-inflammatory cytokines (IL-4, IL-10, and IL-17). Additionally, the percentages of T-cells (CD4^+^, CD8^+^) and dendritic cell phenotypes (CD83^+^, CD86^+^) were significantly elevated (** *p* < 0.01, *** *p* < 0.001) in mice inoculated with the nanovaccine compared to control groups and the rHcES-15 group. Correspondingly, higher levels of antigen-specific serum immunoglobulins (IgG1, IgG2a, IgM) were observed in response to the nanovaccine in comparison to both the antigenic (rHcES-15) and control groups (* *p* < 0.05, ** *p* < 0.01). In conclusion, the data strongly supports the proposal that the encapsulation of rHcES-15 within PLGA NPs effectively triggers immune cells in vivo, ultimately enhancing the antigen-specific adaptive immune responses against *H. contortus.* This finding underscores the promising potential of the nanovaccine, justifying further investigations to definitively ascertain its efficacy.

## 1. Introduction

*Haemonchus contortus* is accountable for significant economic losses due to its voracious blood-sucking behavior, particularly in the abomasum of small ruminants [1]. Infection with this worm results in anemia, diarrhea, weight loss, edema, lethargy, and ultimately death in affected animals [2]. Each mature worm is believed to induce a daily blood loss ranging from 30 to 50 microliters [3]. Anti-parasitic drugs targeting this worm typically encounter resistance from gastrointestinal nematode (GIN) resistance [4]. Barbervax^®^, a vaccine sourced from nematode gut proteins, has been introduced in Australia [1] and shows promise following trials in Brazil [5]. However, its widespread use faces limitations due to its cost, licensing constraints, the need for multiple doses, and the requirement for supplementary strategies like dietary improvements [6]. In the development of a potent vaccine targeting haemonchosis, numerous critical factors must be taken into account. These factors include the inherent diversity within H. contortus, the genetic variability exhibited by the host, the life cycle specificity and distinct stages of the parasite, and the formulation of the vaccine [7]. Thus, there is substantial interest among scientists in developing new approaches, such as selective breeding and vaccinations with nanoparticles, as alternative methods of prevention against *H. contortus* infection [1,8].

During their growth stages, parasites such as *H. contortus* release considerable amounts of excretory/secretory proteins (ESPs) both in controlled laboratory environments and within the bodies of their host animals, notably sheep and goats [3]. These ESPs possess the ability to either circulate in the extracellular space or localize and discharge from the surface of host cells [9]. In comparison to other cellular elements, these proteins are notably sensitive to the effects of drugs [10]. Additionally, due to their capacity to trigger immune responses [11], ESPs offer promising targets for interventions against parasitic infections [12]. Specifically, smaller proteins (ES-15, ES-24) have been isolated and purified from the excretory/secretory proteins (HcESPs) of *H. contortus* in laboratory settings [13]. Furthermore, immunization with ESPs of smaller molecular sizes led to hypersensitivity reactions in genetically resistant sheep [14] and prompted immune responses categorized as Th2 type [15]. Moreover, when interacting with goat peripheral blood mononuclear cells (PBMCs) and dendritic cells (DC), rHcES-15 stimulated the differentiation and proliferation of these cells, inducing significant immunomodulatory functions [16]. This finding offers crucial insights supporting the potential utilization of this antigen in ongoing research endeavors.

In the domain of vaccine development, a strategy involves integrating potent adjuvants to amplify vaccine immunogenicity, thereby eliciting a robust immune response against various pathogens like *Toxoplasmosis* [17] and *Leishmaniasis* [18] in animal models. Without adjuvants, intricate biological molecules such as proteins (antigens) and their components exhibit reduced immunological potency during vaccine formulation [19]. However, the combination of an adjuvant with an antigen has the potential to heighten the immunological response [20]. Studies indicate an 82% enhancement in the efficacy of purified antigens when co-administered with adjuvants in vaccine formulations [21]. Nonetheless, specific adjuvants may induce considerable inflammation, potentially limiting their application in human subjects due to the associated adverse effects [22].

Nanoparticles (NPs) derived from biodegradable and biocompatible polymers, particularly PLGA, have demonstrated both safety and efficacy as adjuvants in drug delivery systems. These NPs excel in encapsulating antigens, thereby facilitating the development of controlled-release NP vaccines that specifically target parasitic infections in murine models [23,24]. Investigations into these polymeric materials in the realm of vaccine delivery have unveiled significant variations in their ability to modulate antigen-specific immune responses [25,26]. Moreover, several studies have observed that the utilization of nano-encapsulation for antigens represents a beneficial strategy in providing immunity in vivo against Anthrax, Tetanus, and Malaria [27].

In our current investigation, we utilized a nanovaccine formulation comprising the PLGA-encapsulated antigen rHcES-15, administered subcutaneously to Institute of Cancer Research (ICR) mice. We conducted a thorough assessment of the resultant immune response induced by the nanovaccine, comparing it meticulously with the response generated by non-encapsulated rHcES-15. Our analysis uncovered escalated secretion levels of cytokines and antibodies, heightened lymphocyte proliferation, and increased expansions of specific immune cells like T cells and dendritic cells (DCs) in reaction to the nanovaccine. This substantiates the immunogenic potential inherent in both the biopolymer PLGA and the antigen itself.

## 2. Materials and Methods

### 2.1. Ethics Declaration

Experimental protocols were executed following the explicit approval obtained from the Science and Technology Agency of the Jiangsu Province (Approval No. SYXK (SU) 2010-0005). All animal procedures rigorously complied with the guidelines outlined by the Animal Welfare Council of China. Stringent efforts were made to minimize potential distress experienced by the animals, with regular health assessments carried out throughout the experimental period.

### 2.2. Animals

Forty female ICR mice, aged 8–10 weeks and weighing between 18–20 g, and Sprague Dawley (SD) rats (Body weight: 140–150 g) were acquired from the Experimental Animal Center of Jiangsu, China PR (SCXK 2017–0001). These mice were maintained in a controlled environment that met specific pathogen-free (SPF) standards, with unlimited access to sterilized food and water.

### 2.3. Optimization of the Working Concentration of Polyvinyl Alcohol (PVA)

In the process of preparing PLGA NPs (PLGA (lactic acid: glycolide 65:35, Molecular weight = 40,000–75,000, Sigma Aldrich, St. Louis, MO, USA)), it is imperative to ascertain the concentration of Polyvinyl Alcohol (PVA) (Molecular weight = 31,000–50,000, Sigma Aldrich, St. Louis, MO, USA) due to its crucial role as a constituent in PLGA [8]. Prior to commencing the preparation of nanoparticles, a comprehensive investigation was conducted to ascertain the most suitable concentration of polyvinyl alcohol (PVA) for optimal performance. Three discrete concentrations of PVA (1%, 4%, and 6%) were employed to determine the ideal working concentration. The resulting PLGA nanoparticles produced under these distinct PVA concentrations were subjected to morphological and size characterization using scanning electron microscopy (JEOL, Akishima-shi, Tokyo, Japan).

### 2.4. PLGA Encapsulation of the rHcES15 Antigen

To synthesize PLGA NPs, the double emulsion method (w/o/w) was employed as described previously [23], with modifications under sterile conditions. Briefly, the rHcES-15 recombinant protein (1 mg/mL, recombinant proteins of HcES-15 and pET-32a expressed in the BL21 (*E. coli*) prokaryotic expression system were obtained from the Laboratory of Molecular Parasitology and Immunology, Nanjing Agricultural University, China PR) [28] was dissolved in a 6% PVA solution to form the inner aqueous phase. Subsequently, a 5% PLGA organic phase in methylene chloride (50 mg PLGA per 1 mL methylene chloride) was prepared. The merging of the inner aqueous phase with the organic phase formed a water-in-oil (w/o) emulsion, achieved through ultrasonic processing (JY92-IIN, Scientz Biotechnology, Ningbo, Zhejiang, China PR) for 4 min (40 w, 5 s, 5 s) in an ice bath. This initial w/o emulsion was then introduced into the aqueous phase containing 6% PVA, and underwent further sonication under the same conditions to yield the ultimate emulsion (water-in-oil-in-water, w/o/w). Additionally, the organic solvent present in the emulsion underwent evaporation under magnetic stirring at 800 revolutions per minute (rpm) for a duration of 4 to 5 h within a fume cabinet at room temperature (RT). Following this process, the nanoparticles (NPs) laden with the antigen were separated from the NP solution through centrifugation at 22,000× *g* times the force of gravity (×*g*) for 45 min at a temperature of 4 °C. Following this, the supernatant was collected for the determination of protein loading efficiency using the Micro-BCA^TM^ protein assay kit (CW Biotech, Beijing, China). The remaining nanoparticles’ precipitate underwent two washes with ultrapure water, underwent freeze-drying for 24 h using Labconco™ equipment from Thermo Fisher Scientific in Waltham, MA, USA, and were then stored at −80 °C for subsequent experiments.

The procedure for manufacturing empty PLGA nanoparticles closely replicated the method employed for creating rHcES-15 loaded PLGA nanoparticles, with the exception of excluding the incorporation of the rHcES-15 protein. Sterile PBS was employed in the initial emulsion formation to generate empty nanoparticles, serving as a negative control for comparative analysis in further experimental investigations.

### 2.5. Characterization of the rHcES-15 Antigen-Loaded PLGA NPs

#### 2.5.1. Encapsulation Efficiency (EE), Loading Capacity (LC), and In Vitro Cumulative Release (CR) of the rHcES-15 Antigen

The post-wash supernatants derived from the nanoparticle purification process were utilized to determine the encapsulation efficiency (EE) and loading capacity (LC) of rHcEs-15, employing the Micro-BCA^TM^ protein assay kit (CW Biotech, Beijing, China), as outlined in previous literature [26,29]. The loading capacity of rHcES15 when combined with PLGA was indirectly assessed by comparing the initial protein quantity used to load the PLGA nanoparticles with the protein content remaining in the supernatant.

The calculations were performed using the following formulas:EE = (total protein − unbound protein)/total protein × 100%
LC = loaded protein/total mass of nanovaccine × 100%

To analyze the antigen’s kinetic profile, we conducted an adjusted in vitro evaluation to measure the cumulative release of rHcES-15 [30,31]. The release of the antigen (rHcES-15) from PLGA nanoparticles was assessed by monitoring alterations in the concentration of free antigens. In summary, lyophilized nanoparticles (3 mg) were dispersed in 150 µL of PBS (pH 7.4, 0.1 M) within a glass container positioned on a shaker bath (37 °C, 120 rpm). Following centrifugation at 12,000× *g* rpm for 15 min, 60 µL of the supernatant was carefully extracted and replaced with an equal volume of fresh PBS at defined time intervals (0, 1, 3, 5, 7, 9, 11, 13, 15, 17, 19, and 21 days).

The concentration of unbound rHcES-15 proteins in the supernatants was quantified using the Micro-BCA^TM^ Protein Assay Kit (CW Biotech, Beijing, China). All experimental procedures were executed in triplicate.

#### 2.5.2. SEM for the Determination of Shape, Size, and Measurement of the Zeta Potential of NPs

The characterization of size and morphology for the rHcES-15+PLGA nanoparticles was performed utilizing a scanning electron microscope (SEM, JEOL IT-100, S-4800 N, Tokyo, Japan). The antigen-loaded nanoparticles were deposited in powdered form onto aluminum stubs that had previously been coated with platinum for analysis. The evaluation of the nanoparticles’ zeta potential involved the utilization of a zeta potential analyzer (Zeta plus, Brookhaven Instruments Co., Holtsville, NY, USA). All assessments of zeta potential were performed under standardized conditions at 25 °C, applying an electric field strength of 11.00 V/cm [32].

#### 2.5.3. SDS-PAGE Analysis

In order to examine the stability of the antigen-loaded nanoparticles (NPs), suspensions containing the antigen underwent a heat-digestion procedure at 95 °C for 20 min. Following this, the digested samples were loaded into a gel matrix at room temperature (RT) for subsequent analysis. Each individual sample was composed of a volume totaling 20 µL, of which 5 µL was designated for the incorporation of molecular weight markers (Thermo Fisher Scientific, Shanghai, China), spanning a range from 10 to 170 kDa. Electrophoretic separation was conducted under constant conditions at 120 volts for a duration of 90 min using the Bio-Rad 300 power pack (Bio-Rad, Hercules, CA, USA). Subsequent to this process, a sodium dodecyl sulfate–polyacrylamide gel electrophoresis (SDS-PAGE) gel was subjected to staining with a 0.025% Coomassie Brilliant Blue solution, facilitating the visualization of distinct antigen bands [33].

### 2.6. Western Blotting

The recombinant HcES-15 protein, post-separation through 12% SDS-PAGE, was transferred onto a polyvinylidene difluoride (PVDF) membrane (Millipore, Burlington, MA, USA) for subsequent Western blot analysis, as detailed in prior work [28]. Following the blockade of non-specific binding with 5% skim milk in Tris-buffered saline containing 0.1% Tween-20 (TBST), the membranes underwent three TBST wash cycles. Subsequently, they were subjected to a 1-h incubation at 37 °C with primary antibodies (anti-rHcES-15) at a 1:100 ratio of dilutions in TBST. After an additional three TBST washes, the membranes were incubated for 1 h at 37 °C with HRP-conjugated rabbit anti-rat IgG (Sigma, USA), diluted at 1:3000 in TBST. Lastly, the visualization of bound antibodies was achieved using a 3,3-diaminobenzidine tetrahydrochloride (DAB) kit (Boster Biotechnology, Wuhan, China) following the manufacturer’s guidelines.

### 2.7. Mice Immunization

Forty ICR mice were randomly distributed into five groups, each comprising eight animals, and received vaccinations on day 0. At the conclusion of the 14-day period, all experimental mice were euthanized using humane methods. The control groups included PBS, pET-32a, and PLGA NP groups, while the experimental (treatment) groups consisted of the rHcES-15 and rHcES-15+PLGA NP (nanovaccine) groups. Subcutaneous injections of a 1 mL vaccine containing 20 μg of the rHcES-15 protein entrapped within PLGA NPs were administered at various sites on the mice, adhering to the previously documented procedure [20]. The details of the vaccination protocol are provided in Appendix A.

### 2.8. Antigen-Specific Serum Antibody Assays

Prior to the scheduled euthanasia on the 14th day, blood samples were collected from the mice in accordance with the method outlined in a prior study [34]. The concentrations of antigen-specific antibodies (IgM, IgG2a, IgG1) within the mouse sera were assessed using mouse ELISA kits sourced from Heng Yuan, Shanghai, China, following the prescribed instructions of the manufacturer. Specifically, 96-well microtiter plates were coated with rHcES-15 at a concentration of 20 µg/mL for the analysis. After triple washing with a solution of PBS (0.01 M) containing 0.05% Tween-20 (PBST), the wells underwent a blocking step using 5% non-fat dry skim milk powder (SMP) dissolved in PBST for a 2-h incubation at 37 °C. Subsequent to this, 100 µL of serum samples (diluted at a 1:50 ratio in PBST-5% SMP) were applied and allowed to incubate for 1 h at 37 °C. Following serum addition, the plates were subjected to HRP-conjugated anti-mouse IgG1, IgG2a, and IgM antibodies (diluted to 1:3000 in blocking buffer (Sigma Aldrich, St. Louis, MO, USA)) for 1 h at 37 °C to determine both antibody levels and isotype distribution. Tetra-methyl benzidine (Sigma-Aldrich, St. Louis, MO, USA) substrate was utilized for color development, and the resultant outcomes were quantified via a spectrophotometry measuring absorbance at 450 nm. All serum samples were evaluated in triplicate.

### 2.9. In Vitro Measurement of Cytokines Using ELISA

Serum samples from different groups of mice were analyzed to quantify the levels of cytokines (IL-4, IL-10, IL-17, IFN-γ, and TGF-β) using ELISA kits acquired from Heng Yuan, Shanghai, China PR. The assay procedures were carried out in accordance with the manufacturer’s specified protocols.

### 2.10. Evaluation of Splenic Lymphocyte Proliferation Assay

To evaluate the proliferative potential of splenic lymphocytes across all mice, a lymphocytic proliferation assay was executed [35]. On the 14th day, the mice were humanely euthanized to aseptically isolate spleen lymphocytes and antigen-presenting cells (APC) using the Mouse Spleen Lymphocyte Isolation Kit (TBD, Tianjin, China). The cellular concentration was adjusted to 1 × 10^7^ cells/mL in an RPMI-1640 culture medium (Gibco, Carlsbad, CA, USA) and incubated overnight in 6-well cell culture plates. Subsequently, cell supernatants, containing T and B cells, were harvested, their concentrations were modified to 1 × 10^7^ cells/mL, and then 1 × 10^6^ cells in 100 µL of culture medium (CM), supplemented with 10% heat-inactivated fetal bovine serum (FBS), 100 U/mL of penicillin, and 100 mg/mL of streptomycin (Gibco, Carlsbad, CA, USA), were added to individual wells of round-bottom 96-well culture plates. Following this, the cells were stimulated with rHcES-15 at a concentration of 4 µg/mL and cultured for an extra 72-h duration under conditions of 5% CO_2_ at 37 °C. In addition, control samples treated with concanavalin A (ConA, Sigma Aldrich, St. Louis, MO, USA) were used, with one group containing rHcES-15 and another without, serving as positive and blank controls, respectively [25]. The investigation of the lymphocyte proliferation stimulated by the antigen (rHcES-15) within the collected cells was conducted using the Enhanced Cell Counting Kit-8 in accordance with the manufacturer’s instructions (CCK-8, Beyotime, Shanghai, China). The optical density at a specific wavelength of 450 nm (A450 value) was assessed through a microtiter ELISA reader (Thermo Scientific Multiskan FC, Waltham, MA, USA). The findings were represented in terms of the stimulation index (SI), calculated by employing the prescribed equation [24]:SI (%) = At/Ac × 100
where At represents the mean A450 value of the specific test group, and Ac represents the mean A450 value of the blank control group.

### 2.11. Analysis of Lymphocyte Phenotypes via Flow Cytometry

Flow cytometry, in accordance with previously established protocols [24] was utilized to assess the distribution of CD4^+^ and CD8^+^ T cells within the spleens of all subjects in the experimental cohort. The procedure for isolating T and B cells from the immunized mice mirrored the technique employed in the splenic lymphocyte proliferation assay. To determine the proportions of CD4^+^ and CD8^+^ T cells, cells were specifically labeled using Hamster anti-CD3e-APC, Rat anti-CD4-PE, Hamster anti-CD3e-APC, and Rat anti-CD8a-PE antibodies obtained from Bio-legend, San Diego, CA, USA. Following labeling, the stained cells underwent analysis using a fluorescence-activated cell sorting machine manufactured by BD Biosciences located in Franklin Lakes, NJ, USA.

### 2.12. Determination of DC Phenotypes via Flow Cytometry

The spleens were harvested from the experimental mice, and the splenic cells were isolated under sterile conditions using the Spleen Lymphocytes Isolation Kit. Subsequently, the splenic lymphocytes were cultured overnight in RPMI-1640 medium within 6-well culture plates. The supernatant, consisting of non-adherent cells, was removed, and the cell plate wells were rinsed three times with PBS to eliminate any residual non-adherent cells. The adherent cells were gently detached using pipetting methods. Following the centrifugation and additional washing steps, the cells were labeled with Hamster anti-CD11c-APC, Rat anti-CD83-PE, Hamster anti-CD11c-APC, and Rat anti-CD86-PE antibodies to evaluate the proportions of CD83^+^ and CD86^+^ cells on the dendritic cells (Bio-legend, San Diego, CA, USA). Subsequent analysis involved flow cytometry conducted using FACS Caliber equipment (BD Biosciences, Franklin Lakes, NJ, USA).

### 2.13. Statistical Analysis

This experimental procedure underwent three repetitions, and the resultant data were represented as mean values accompanied by the standard error of the mean (SEM). Group distinctions were assessed through one-way analysis of variance (ANOVA) followed by Dunnett’s multiple comparison test, facilitating the comparison of treatment means against the mean of the control group. Statistical significance was denoted by symbols: * for *p*-values < 0.05, ** for *p*-values < 0.01, and *** for *p*-values < 0.001. An analysis of the FACS data was executed using Flow Jo version 10 software (version 10, Franklin Lakes, NJ, USA).

## 3. Results

### 3.1. Determination of Optimum PVA Concentration

SEM analysis was utilized to investigate the structural morphology and size parameters of the PLGA NPs manufactured with varying concentrations of PVA (1%, 4%, and 6%). The findings revealed that NPs formulated with 6% PVA exhibited a smaller and more uniform particle size, measuring approximately 350 nm (Figure 1A). Conversely, PLGA NPs prepared using 1% and 4% PVA displayed larger and less uniform particle sizes and shapes in comparison to those synthesized with 6% PVA (Appendix A). Consequently, among the tested PVA concentrations, 6% PVA was identified as the most suitable choice for subsequent experimental investigations.

### 3.2. The Characterization of Antigen-Loaded NPs

Based on the SEM analysis, the nanoparticles exhibited a smooth morphology and displayed a consistently uniform size range of 350–400 nm (Figure 1A,B) when formulated with 6% PVA. Additionally, the zeta potential of these nanoparticles was measured at 35 ± 1.9 mV (Figure 1C).

The integrity of rHcES-15 was evaluated using SDS-PAGE with a 12% separating gel, both in the presence and absence of PLGA NPs. The results distinctly indicate that the process of formulating nanoparticles did not cause any alteration in the molecular weight (Mw) of rHcES-15, as evidenced by the bands observed at approximately 33 kDa in size (Figure 1D).

Furthermore, an in vitro cumulative release assay (Figure 1G) was conducted to investigate the kinetic profile of the rHcES-15 antigen. The data unveiled an approximately 72% release of the antigen from the PLGA after a 21-day period. These findings suggest the potential of nanoparticles to effectively deliver the antigen, showcasing stability and maintaining small particle sizes.

### 3.3. Immuno-Blot Analysis

The Western blot analysis demonstrated the specific recognition of the recombinant HcES-15 protein via the immune sera, particularly rat anti-rHcES-15, manifesting as a well-defined band at approximately 33 kDa. In contrast, the sera obtained from normal rats did not display any discernible recognition of the aforementioned protein (Figure 1E,F).

### 3.4. Evaluation of Serum Antibody Levels Induced by the Nanaovaccine

In this experiment, the concentrations of IgM, IgG1, and IgG2a antibodies in sera collected from distinct mice were quantified using ELISA, as depicted in Figure 2. Notably, the IgG1 levels in both the rHcES-15 and rHcES-15+PLGA NPs groups were significantly higher compared to the PBS (blank control), pET-32a, and notably higher than the PLGA NPs groups alone (* *p* < 0.05, ** *p* < 0.01). Additionally, mice administered with the antigen (rHcES-15) and the nanovaccine (rHcES-15+PLGA NPs) displayed significantly higher levels of IgG2a and IgM compared to the PBS, pET-32a, and PLGA NPs groups (* *p* < 0.05, ** *p* < 0.01).

Furthermore, the nanovaccine demonstrated a notably heightened production of immunoglobulins (IgG1, IgM) compared to the rHcES-15 alone (* *p* < 0.05) (Figure 2A,C).

### 3.5. Cytokines Secreted by the rHcES-15 Antigen and Nanovaccine

As per the results, the serum samples from mice subjected to immunization with rHcES-15 and rHcES-15+PLGA NPs showcased notably increased concentrations of IL-4 and IL-10 in comparison to the control groups (* *p* < 0.05, *** *p* < 0.001). Furthermore, a significant elevation in the levels of both cytokines was evident in the rHcES-15+PLGA NPs group when contrasted with the rHcES-15 group (* *p* < 0.05) (Figure 3A,B).

Compared to the PBS group, both the rHcES-15 and rHcES-15+PLGA NPs groups showed significant increases in IL-17 levels (*** *p* < 0.001). Furthermore, IL-17 secretions were significantly enhanced in these groups compared to the pET-32a and PLGA NPs groups (* *p* < 0.05, ** *p* < 0.01). Moreover, the group of mice that received the nanovaccine induced more IL-17 (* *p* < 0.05) than the rHcES-15 group (Figure 3C).

A notable decrease in TGF-β production (* *p* < 0.05) was evident in both treatment cohorts (rHcES-15 and rHcES-15+PLGA NPs) when compared to the PBS, pET-32a, and PLGA NPs groups (Figure 3D). Similarly, a comparatively lower, yet statistically nonsignificant, secretion of IFN-γ was observed in the experimental groups in contrast to the control groups (Figure 3E).

### 3.6. Nanovaccine-Induced Splenic Lymphocytes Proliferation

Splenic lymphocytes isolated from all experimental mice were subjected to evaluation for their proliferative responses, particularly in response to the rHcES-15, as illustrated in Figure 4. The stimulation index (SI) revealed substantial proliferation in the positive control (ConA), rHcES-15+PLGA NPs, and rHcES-15 groups compared to the PBS, pET-32a, and PLGA NPs groups (* *p* < 0.05, *** *p* < 0.001). Notably, the positive control (ConA) exhibited the highest proliferative response in comparison to both the rHcES-15 and rHcES-15+PLGA NPs groups.

### 3.7. The Nanovaccine Elicited the Activation of CD4^+^ and CD8^+^ T Cells

Utilizing flow cytometry, the assessment of CD4^+^ and CD8^+^ T cell proportions within lymphocyte subsets was conducted across each group (Figure 5). Substantial elevations in the percentages of CD3e^+^CD4^+^ and CD3e^+^CD8a^+^ cells were observed in the treatment groups in comparison to the PBS, pET-32a, and PLGA NPs groups (** *p* < 0.01, *** *p* < 0.001). Specifically, the rHcES-15+PLGA NPs group displayed a discernible rise (* *p* < 0.05) in CD3e^+^CD4^+^ cells relative to the rHcES-15 group (Figure 5A,C). Notably, the CD4^+^ values in the PLGA NPs group were higher than those in the PBS and pET32a groups, potentially attributable to their adjuvant effects, although remaining lower than those elicited by the rHcES-15 antigen and the nanovaccine (rHcES-15+PLGA NPs group). Moreover, mice immunized with the nanovaccine (rHcES-15+PLGA NPs) exhibited increased CD8a^+^ levels (* *p* < 0.05) compared to the rHcES-15 group (Figure 5B,D).

### 3.8. The Nanovaccine Increased the Activation and Maturation of DC Phenotypes

Flow cytometry was employed to investigate the interaction between antigen-loaded nanoparticles (NPs) and dendritic cells (DCs) while assessing the alterations in DC phenotypes. The percentages of CD83^+^ and CD86^+^ cells were determined across various mouse groups (Figure 6). The findings unveiled substantial elevations in the percentage of CD11c^+^CD83^+^ and CD11c^+^CD86^+^ cells in both the rHcES-15 and rHcES-15+PLGA NPs groups when compared to the PBS, pET-32a, and PLGA NPs groups (* *p* < 0.05, ** *p* < 0.01, *** *p* < 0.001). Moreover, the rHcES-15+PLGA NPs group exhibited notably higher proportions of CD11c^+^CD83^+^ cells in contrast to the rHcES-15 group (** *p* < 0.01).

## 4. Discussion

In regions characterized by warmth and humidity, the prevalence of *H. contortus* infection presents a substantial challenge to domesticated livestock. This nematode actively releases a wide variety of excretory/secretory proteins (ESPs) into the host environment, capable of manipulating the host’s immune system by either modulating or suppressing its functions [3]. A subset of these HcESPs comprises low molecular weight ES antigens, which are crucial in directing a specific immune response against this helminth infection within the host [13]. Typically, purified proteins demonstrate limited immunogenicity, necessitating the assistance of an effective adjuvant to induce a robust immune reaction [36]. In the current study, the western blot assays confirm that the protein was completely purified and has specificity with the polyclonal antibody produced in the SD rats [28]. Additionally, we utilized biodegradable polymers in the form of PLGA NPs as an adjuvant to transport the rHcES-15, aiming to explore its potential in eliciting both Th1 and Th2 immune responses against *haemonchosis*. Mice inoculated with antigen-loaded NPs exhibited heightened levels of antigen-specific antibodies, elevated cytokine concentrations in sera, increased proportions of T cells and DCs, and augmented lymphocyte proliferation [26]. These findings strongly advocate the combined use of the recombinant antigen of *H. contortus* (rHcES-15) with PLGA NPs, showcasing promising prospects for therapeutic interventions.

The primary drawback associated with PLGA-based NPs revolves around their limited loading capacity (LC), indicating the ratio of the loaded drug or antigen quantity to the total NP quantity. Despite generally exhibiting high encapsulation efficiency (EE), achieving substantial drug loading remains a challenge [37]. In our investigation, we noted an LC of 25 ± 1.1 alongside a notably high EE of 72.37 ± 3.51 for PLGA NPs (Table 1). Current scientific literature underscores that the optimal NP size for effective uptake by dendritic cells (DCs) should not exceed 500 nm [38]. Consequently, PLGA NPs measuring 350 nm demonstrated superior internalization and activation of DCs compared to larger particles [39]. In line with earlier studies, we identified the appropriate size of rHcES-15+PLGA NPs conducive to efficient uptake by DCs.

In a recent study, mice immunized with NPs carrying the antigen demonstrated heightened antibody levels and a potent lymphoproliferative response compared to control groups and the antigen administered alone (rHcES-15). The elevated levels of antigen-specific IgG subtypes correlated with the host’s defense against *H. contortus* challenge [40]. Similar findings have been reported in prior experiments involving different nematodes [41,42]. These results suggest the potential of this recombinant protein as a viable candidate for a nanovaccine targeting *H. contortus* infection. Previous research has highlighted the multifaceted roles of ESPs derived from various helminths, primarily influencing the development of host immune cells [43,44]. In our current investigation, we observed a robust lymphoproliferative response induced by both the rHcES-15 alone and with the nanovaccine (rHcES-15+PLGA NP), as illustrated in Figure 4. This phenomenon might be linked to an upsurge in IgG1 production.

The pivotal role of IgM in triggering the immune response during parasitic infections is recognized. It functions as a crucial mediator linking the innate and adaptive immune systems, enhancing specific immune reactions, and controlling the proliferation of various pathogens, such as *Vesicular stomatitis* virus, *Lymphocyte choriomeningitis* virus, *Vaccinia* virus, *Listeria monocytogenes*, and systemic bacterial infections. Concurrently, employing passive immunization techniques utilizing the hyperimmune fractions that are abundant in IgG2a antibodies may potentially induce prolonged delays in parasite presence within the host’s system [45]. In the current investigation, significantly elevated levels of both IgM and IgG2a were observed in the serum of mice injected with the antigen and nanovaccine compared to the control groups (* *p* < 0.05, ** *p* < 0.01).

Precise signaling molecules play a crucial role in governing intercellular communication within the immune system. These entities, specifically cytokines and chemokines, are pivotal in coordinating immune responses [46]. For instance, the IFN-γ released by Th1 cells regulates cellular immune responses against microbial invasions and influences Th1/Th2 cell differentiation. Conversely, IL-4 dictates the type of immunity and pathogenesis that occurs during nematode invasions [47]. Notably, during *H. contortus* infection, the prevailing secretion of IL-4 aligns with a Th2 type of immune response [48]. The anti-inflammatory cytokine IL-10, recognized as a master regulator, mitigates immune responses and dampens the tissue damage caused by inflammation, which is critical for effective host defense [49]. In line with previous studies of rHcES-15 [28], our research suggests that rHcES-15 exhibits a moderate inhibition of IFN-γ secretion, contributing to maintaining a balanced Th1/Th2 environment during the host–parasite interaction. Moreover, mice vaccinated with the nanovaccine display heightened secretions of IL-4 and IL-10, indicating the need for further investigation and discussion.

Th17 cells assume a critical role by producing the IL-17 cytokine, which significantly influences the pathogenesis of various helminths, acting as a regulator of tissue inflammation [50]. Remarkably, an evident elevation in IL-17 secretion was observed in goat peripheral blood mononuclear cells (PBMCs) when exposed to rHcES-15 in vitro [28]. TGF-β serves as a pivotal regulator in controlling diverse cellular activities encompassing cell proliferation, growth, differentiation, pro-inflammatory responses, and various immunomodulatory functions [51,52]. The secretion levels of both TGF-β and IL-10 are subject to the influence of factors such as the severity of infection, parasite developmental stage, and host genetic factors [53]. In our present investigation, a notable reduction was discerned in the secretion of TGF-β within our experimental groups (rHcES-15, rHcES-15+PLGA NPs). These results indicate that the cellular immune response induced by this antigen predominantly involves a combination of Th1 and Th2 immune responses.

Lymphocyte proliferation constitutes a fundamental aspect within immunological investigations [54]. In our current study, we conducted an assessment of lymphocyte proliferation across all experimental groups. Our results demonstrated notably elevated proliferations following exposure to rHcES-15 and rHcES-15+PLGA compared to empty NPs, pET32a, and PBS. This serves as further substantiation that the encapsulation of this antigen within nanoparticles possesses the capability to augment the host animal’s immune response against *H. contortus*.

As evidenced in this investigation, the concurrent application of the helminth antigen (rHcES-15) alongside polymeric NPs resulted in the elevation of effector cytokines, likely due to the enhanced MHC molecule presentation facilitated by dendritic cells (DCs). Activation of DCs serves as a pivotal trigger for priming both CD4^+^ and CD8^+^ T cells [55]. Prior research has demonstrated that liposome-triggered stimulation notably induced the cross-priming of CD8a^+^ T cells in mice [56], and enhanced the survival of CD4^+^ T cells [57]. Consistent with these observations, our findings unveiled a more robust proliferation of activated T cells in the immunized mice compared to control groups (Figure 5). Previous studies have also indicated the significant proliferation of CD4^+^ and CD8^+^ T cells upon exposure to *H. contortus* antigens in host animals [14,26,40]. Collectively, these outcomes imply that the combination of rHcES-15 with the adjuvant effect of PLGA NPs substantially contributes to stimulating both cellular and humoral immunity in vivo.

Dendritic cells (DCs) are essential for antigen presentation and the initiation of the adaptive immune response following vaccination [58]. Their primary role involves presenting antigens as foreign entities, processing them during DC maturation, and presenting antigen peptides to CD4^+^ or CD8^+^ T cells via the MHC-II or MHC-I pathways, respectively [59]. Our study investigated the properties of rHcES-15 and PLGA NPs using splenic DCs (CD11c^+^CD83^+^, CD11c^+^CD86^+^). Previous research has highlighted the significant impact of rHcEs-15 on the maturation and differentiation of monocyte-derived DCs in goats [16]. As depicted in Figure 6, administering the nanovaccine effectively stimulated DCs in mice, demonstrating a marked distinction compared to the controls and the antigen (rHcES-15) groups. Therefore, the combined use of PLGA NPs and rHcES-15 might represent a more effective vaccination strategy for animals than using the antigen or PLGA alone.

## 5. Conclusions

Our findings provide clear evidence that immunization using this nanovaccine (rHcES-15+PLGA NPs) effectively triggers robust immune responses against *H. contortus* infection. We observed a significant upsurge in the specific antibody and cytokine levels in the sera of immunized mice. This nanovaccine notably augmented the proportions of T cells, facilitated dendritic cell (DC) maturation, and stimulated considerable lymphocyte proliferation. Notably, these results underscore the ability of this nanovaccine to induce a markedly enhanced immune response compared to the use of the rHcES-15 antigen alone.

## Figures and Tables

**Figure 1 vaccines-11-01794-f001:**
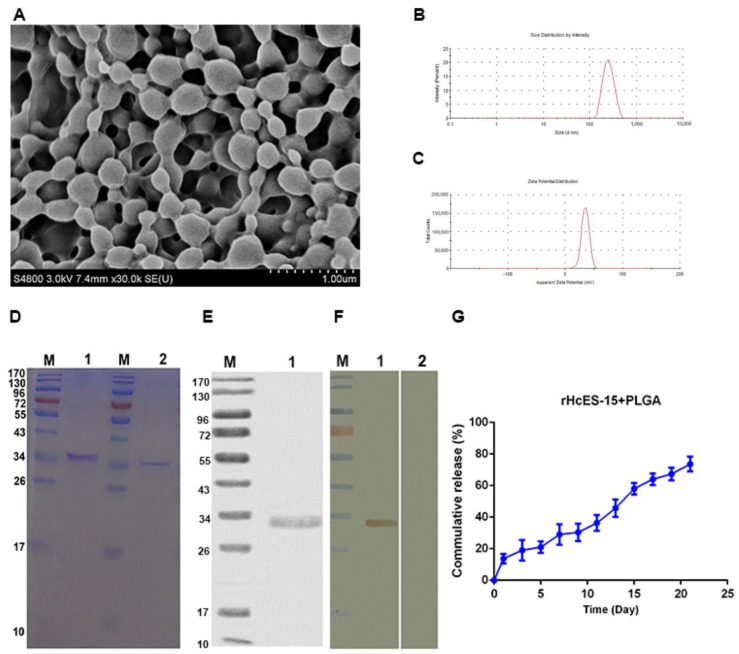
This study extensively investigated the physical properties and structure of antigen-loaded nanoparticles (NPs) using observations at a magnification of 10,000×. This encompassed a detailed analysis of antigen integrity following its integration into the nanoparticles, purification methodologies, Western blot assays, and the cumulative release kinetics. The findings were systematically categorized into several sections: Subsection (**A**) showcased scanning electron microscopy (SEM) images, offering visual insights into the morphology of rHcES-15+PLGA NPs containing 6% PVA. Subsection (**B**) presented the distribution of sizes for the antigen-loaded NPs. Subsection (**C**) illustrated the zeta potential, a critical measure of surface charge, of these NPs. Subsection (**D**) demonstrated sodium dodecyl sulfate–polyacrylamide gel electrophoresis (SDS-PAGE) outcomes that confirmed the antigen’s integrity post-incorporation into the polymeric matrix. This analysis included standard molecular weight markers, pure rHcES-15, and rHcES-15 attached to PLGA NPs. Subsections (**E**,**F**) explored the purification strategy involving chromatography and Western blotting for validation of the recombinant protein (rHcES-15). A discrepancy in observed molecular weight (33 kDa) was attributed to an additional fused vector protein (18 kDa). Deducting the vector protein’s size verified the anticipated 15 kDa size for rHcES-15. The Western blot analysis further confirmed the presence of the purified rHcES-15 protein via specific probing with rat anti-rHcES-15 sera. Subsection (**G**) presented the release profile of the antigen from PLGA NPs under physiological conditions (pH 7.4, 37 °C) over 21 days, demonstrating the percentage of antigen released over time.

**Figure 2 vaccines-11-01794-f002:**
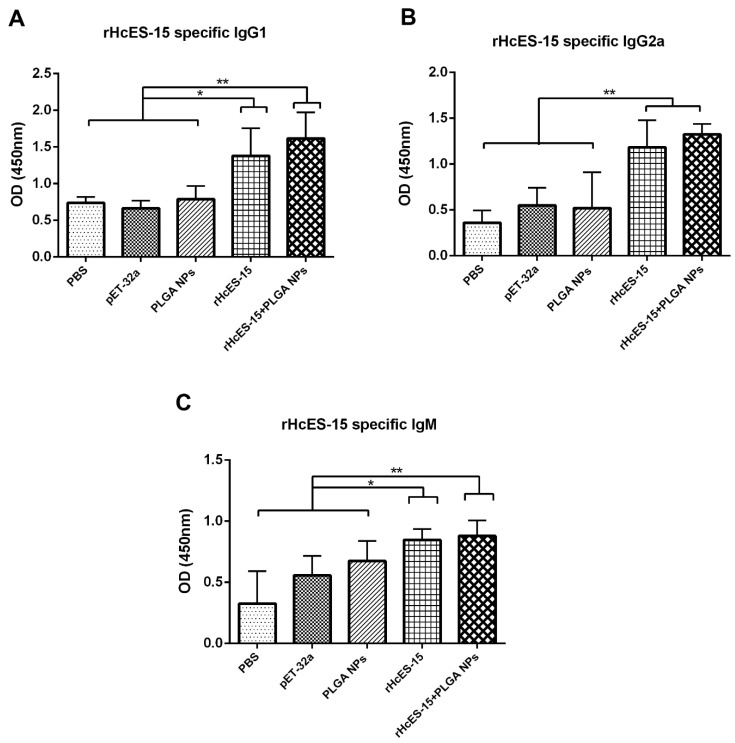
The influence of both the antigen and antigen-loaded nanoparticles (NPs) on serum IgG1 (**A**), IgG2a (**B**), IgM (**C**) in mice was assessed using ELISA. The presented data are a representative of triplicate experiments, denoted as statistical significance at levels * *p* < 0.05 and ** *p* < 0.01.

**Figure 3 vaccines-11-01794-f003:**
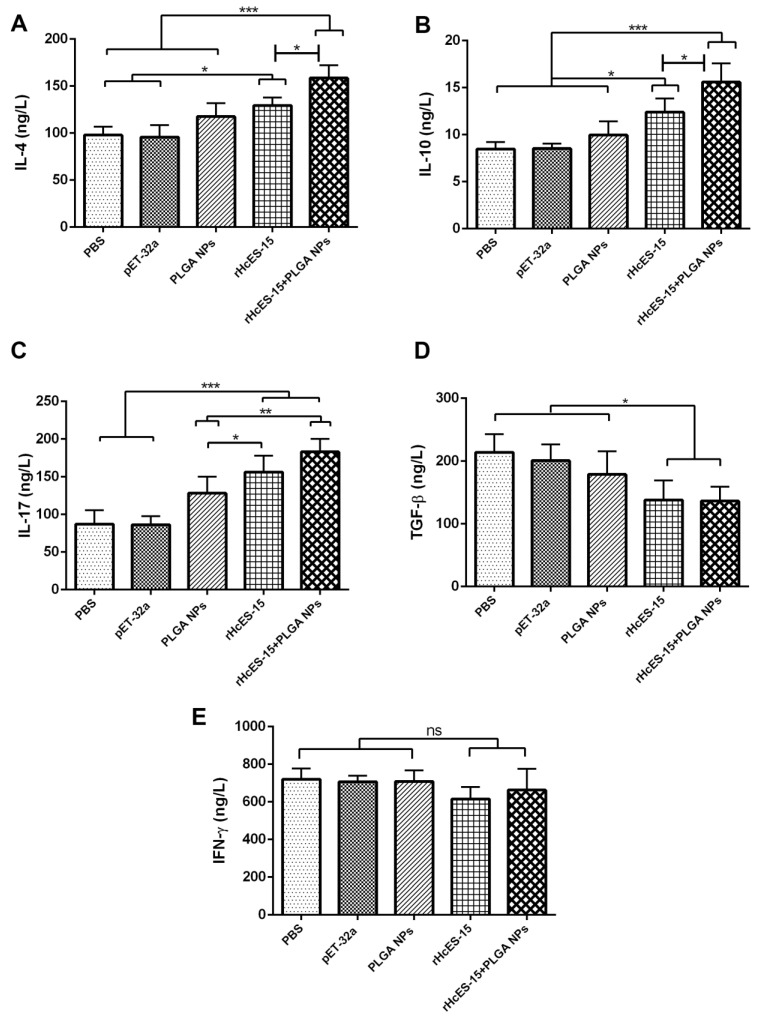
Cytokine expression levels were ascertained via ELISA in the sera of mice following vaccination with distinct antigen delivery systems. Eight mice (n = 8) received a single immunization with the antigen and nanovaccine on day 0. The concentrations of IL-4 (**A**), IL-10 (**B**), IL-17 (**C**), IFN-γ (**D**), and TGF-β (**E**) were shown respectively. The presented data is a representative of independent triplicate experiments, with statistical significance indicated as * *p* < 0.05, ** *p* < 0.01, and *** *p* < 0.001. “ns” means non-significant.

**Figure 4 vaccines-11-01794-f004:**
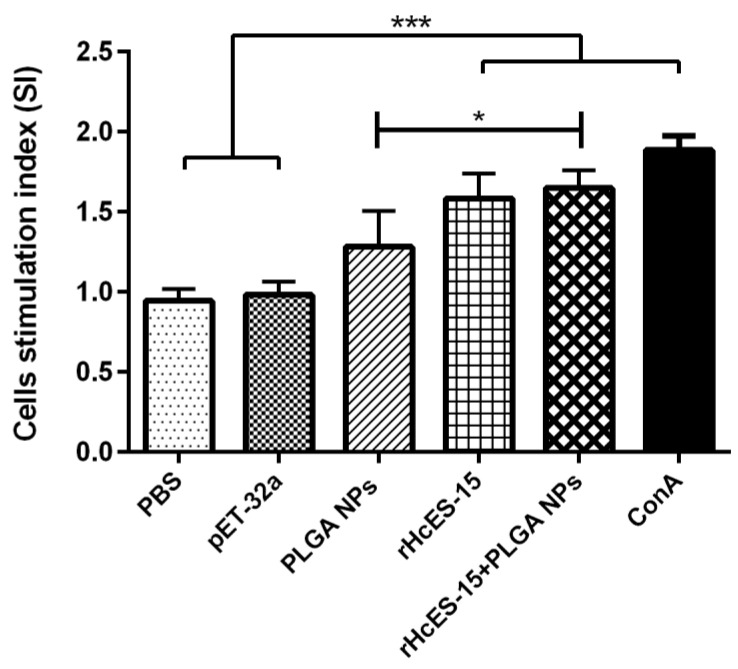
The proliferation index of splenocytes from different immunized mice was assessed following in vitro stimulation with various treatments. The data provided are an aggregate of three independent experiments, and the presented values indicate the means ± SEM (* *p* < 0.05, *** *p* < 0.001).

**Figure 5 vaccines-11-01794-f005:**
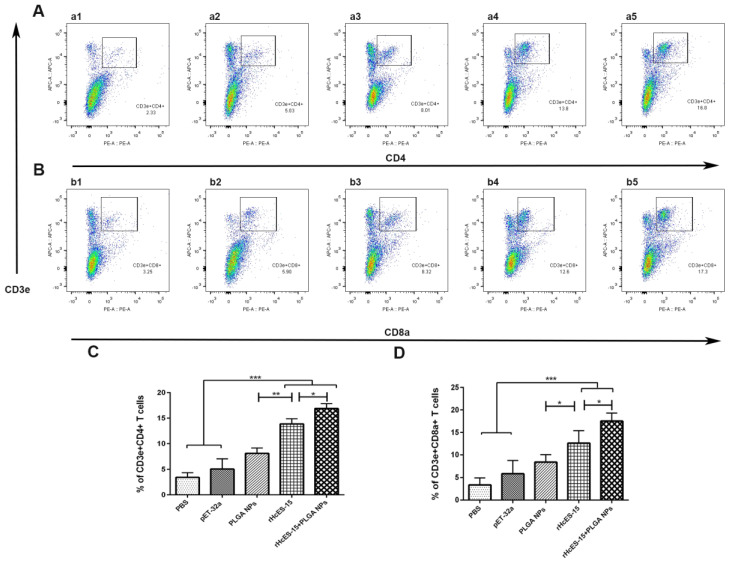
The alterations in the proportions of CD4^+^ and CD8^+^ T cells within different mouse groups were quantified through flow cytometry analysis. Figure 5 illustrates the impact of distinct treatments on CD4^+^ and CD8^+^ T cell proportions (**A**,**B**). In panels A and B, a1 and b1 represent the PBS group (blank control), a2 and b2 depict the pET-32a vector protein group, a3 and b3 represent the PLGA NPs group, a4 and b4 represent the rHcES-15 group, and a5 and b5 represent the rHcES-15+PLGA NPs group. The graphical representation of CD4^+^ and CD8^+^ T cell percentages across all groups is depicted in Figure 5, panels (**C**,**D**). The findings presented here are derived from a singular experiment, indicative of three independent experiments (* *p* < 0.05, ** *p* < 0.01, *** *p* < 0.001).

**Figure 6 vaccines-11-01794-f006:**
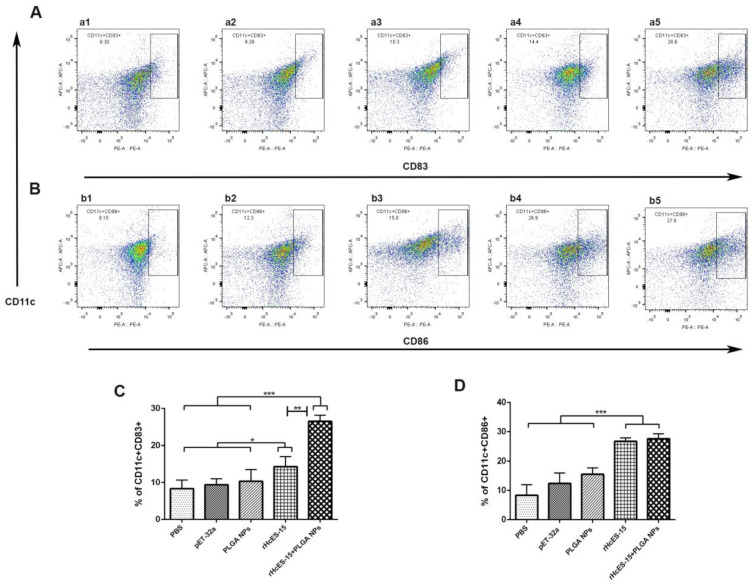
Flow cytometry analysis was utilized to investigate how different antigen delivery systems influence maturation and antigen presentation in splenic dendritic cells. The levels of CD11c^+^CD83^+^ and CD11c^+^CD86^+^ expression in splenic dendritic cells were assessed across five groups (**A**,**B**). In panels (**A**,**B**), a1 and b1 represent the PBS group (blank control), a2 and b2 depict the pET-32a vector protein group, a3 and b3 represent the PLGA NPs group, a4 and b4 represent the rHcES-15 group, and a5 and b5 represent the rHcES-15+PLGA NPs group. The data presented in (**C**,**D**) showed the percentages of CD11c^+^CD83^+^ and CD11c^+^CD86^+^ cells as the mean ± SEM of triplicate experiments (* *p* < 0.05, ** *p* < 0.01, *** *p* < 0.001).

**Table 1 vaccines-11-01794-t001:** Characterization of Recombinant Antigen (rHcES-15)-loaded PLGA NPs. Data are presented as the mean ± SD (n = 3).

Antigen + NPs	Size (nm)	LC^a^ (%)	EE^b^ (%)	Zeta Potential (mV)
rHcES-15+PLGA NPs	350 ± 40	25 ± 1.1	72.37 ± 3.51	35 ± 1.9

LC^a^ = (total protein − unbound protein)/total dry weight of Nanovaccine × 100%. EE^b^ = (total protein − unbound protein)/total protein × 100%.

## Data Availability

Data is contained within the article.

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
