# Peer review of "PLGA-Encapsulated Haemonchus contortus Antigen ES-15 Augments Immune Responses in a Murine Model"

_vaccines, 2023, doi:10.3390/vaccines11121794_

Round 1

Reviewer 1 Report

Comments and Suggestions for Authors

The manuscript is interesting for readership and it is acceptable after a small modification. 

1. Page 4, line 186: “0 days” or Table 1 “0 Day” should be “Day 0” 2. The expressions of significance levels in Figures 2, 3 and their interpretations in sections 3.4 and 3.5 are difficult to understand.

Author Response

The manuscript is interesting for readership and it is acceptable after a small modification. 

Comments 1: Page 4, line 186: “0 days” or Table 1 “0 Day” should be “Day 0”

Author response: We appreciate for your valuable comments. The changes have done in main text file.

Comment 2: The expressions of significance levels in Figures 2, 3 and their interpretations in sections 3.4 and 3.5 are difficult to understand.

Author response: Thank you very much for your comments. Figure 2 and 3 are changed in the revised version of the manuscript.

Reviewer 2 Report

Comments and Suggestions for Authors

The manuscript entitled "PLGA Encapsulated Antigen ES-15 of Haemonchus contortus Improves the Immune Responses in Murine Model” has been written well and the experimental design is optimal for determining PLGA encapsulated treatment substance. However, The author must improve the following points to improve the manuscript further:

Page 8, Line 289, 298: Figure E: Author must describe in vitro release procedure in detail in material and method section. Author must add method and specify in which matrix or media release profile has been determined.

Line 303: Monitoring Clinical Signs and Local Reactions Caused by the Nanovaccine should be monitored using hematological and biochemical tests. However, the manuscript lacks that data, which needs to be provided.

Line 324: Why only a few cytokines have been monitored? The author must determine other strong cytokines like IL-1Beta, IL-6, and INF-Gamma to evaluate its true effect.

Author Response

The manuscript entitled "PLGA Encapsulated Antigen ES-15 of Haemonchus contortus Improves the Immune Responses in Murine Model” has been written well and the experimental design is optimal for determining PLGA encapsulated treatment substance. However, the author must improve the following points to improve the manuscript further:

Comment 1: Page 8, Line 289, 298: Figure E: Author must describe in vitro release procedure in detail in material and method section. Author must add method and specify in which matrix or media release profile has been determined.

Author response: Thank you very much for your valuable suggestion. This experiment was conducted in PBS and the details are included in the main text file.

Comment 2: Line 303: Monitoring Clinical Signs and Local Reactions Caused by the Nanovaccine should be monitored using hematological and biochemical tests. However, the manuscript lacks that data, which needs to be provided.

Author response: We highly appreciate your comments. Hematological and biochemical methods were not employed to assess the clinical signs and local reactions; rather, these were observed through gross examination and daily lesion assessments. Consequently, the data pertaining to these observations could not be obtained. Therefore, it is suggested to remove the sections detailing the monitoring of Clinical Signs and Local Reactions Caused by the Nanovaccine (2.8, 3.3) from the text file.

Comment 3: Line 324: Why only a few cytokines have been monitored? The author must determine other strong cytokines like IL-1Beta, IL-6, and INF-Gamma to evaluate its true effect.

Author response: The current investigation represents an initial trial in our pursuit to develop a nanovaccine specifically aimed at the final host, (which is Goat), of the nematode Haemonchus contortus. The assessment of the aforementioned cytokines will be conducted and documented in our subsequent publication. We express our gratitude to the reviewer for bringing forth this insightful suggestion.

Reviewer 3 Report

Comments and Suggestions for Authors

Comments -

6% PVA displayed a relatively smaller particle size, typically around 350 nm - why and how? Need justification. 

Why PLGA NPs - why and how? Need justification. Since various biodegradable and biocompatible polymers are available. 

Fig. 1 A SEM - particle not look like uniform. Need more detailed scanning for for proper results. 

Fig. 1 E - Kindly put graph mean ± standard error of the mean 

Comments on the Quality of English Language

NA

Author Response

Comment 1: 6% PVA displayed a relatively smaller particle size, typically around 350 nm - why and how? Need justification. 

Author response: Thanks for your comments. Actually, PVA exhibits physical properties similar to biological tissues, fostering compatibility. Its biocompatibility arises from a molecular structure conducive to protein absorption, minimal cell adhesion, and absence of toxic effects. As a result, PVA membranes have gained prominence in biomedical applications [1]. Additionally, PVA demonstrates the capability to establish chemical bonds or physical entanglements with nanoparticle surfaces [2].

Incorporating 6% PVA (polyvinyl alcohol) into the formation of PLGA (poly(lactic-co-glycolic acid)) nanoparticles has consistently demonstrated a correlation with the generation of smaller particle sizes. The presence of PVA in the process, acting as a stabilizer or surfactant, plays a crucial role in controlling aggregation and size regulation during nanoparticle synthesis.

This trend towards smaller PLGA nanoparticle sizes facilitated by 6% PVA stems from several contributing factors. Specifically, PVA's ability to stabilize the PLGA solution prevents particle aggregation and coalescence, while also reducing interfacial tension, leading to the formation of smaller, more uniformly sized particles. The concentration of 6% PVA seems to strike an optimal balance between stabilizing effects and surface tension reduction, influencing the nucleation and growth kinetics of the nanoparticles, resulting in their smaller size distribution.

Overall, the inclusion of 6% PVA in the process of generating PLGA nanoparticles serves as an effective means to obtain smaller particle sizes, enhancing uniformity and control during nanoparticle fabrication for various biomedical and pharmaceutical applications.

Comment 2: Why PLGA NPs - why and how? Need justification. Since various biodegradable and biocompatible polymers are available. 

Author response: Thank you for your valuable comment. In fact, there are two reasons to use PLGA in the current investigation.

  1. Various other researches have used PLGA in the immunogenic studies, and it has been already documented that PLGA containing the adjuvant properties also. the overall results of PLGA nanoparticles were better than chitosan which is also a biodegradable nano-material. PLGA (poly(lactic-co-glycolic acid)) stands out among biodegradable polymers due to its exceptional biocompatibility, controllable degradation, versatility in various forms, FDA approval for medical use, stability, and precise drug release capabilities. Its tunable degradation rate allows tailored release kinetics, making it valuable in drug delivery and tissue engineering. FDA-approved, safe, and with minimal toxicity, PLGA maintains stability, ensuring prolonged drug integrity until targeted release. Upon breakdown, it produces harmless by-products and its controlled drug release reduces dosing frequency, enhancing patient compliance.
  2. PLGA is used in our lab as it provided a great encapsulation of antigens we used frequently. Moreover, it has stimulated the antigen presenting cells as well as T cells [3–5].

Comment 3: Fig. 1 A SEM - particle not look like uniform. Need more detailed scanning for proper results. 

Author response: Thank you very much for your valuable suggestion. This figure of PLGA NPs is already attached with the main text as supplementary file showing that the particle size and the shape of NPs were not same with each other when 1 and 4 % PVA were used in the formulation of PLGA NP. As compared to these concentration, the particle assize as well as, the shape of NPs was alike with 6% PVA concentration. We will consider more detailed study in our next publication.

Figure S1: Scanning electron microscopic images of PLGA NPs at different concentrations of PVA. The scale bar is 2 μm.

Notes: (A) 1 % PVA. (B) 4 % PVA.

Abbreviations: NPs, nanoparticles; PVA, polyvinyl alcohol.

Comment 4: Fig. 1 E - Kindly put graph mean ± standard error of the mean 

Author response: We are highly thankful for your suggestion. The figure is revised in the main text file.

  1. Yang, J.M.; Su, W.Y.; Leu, T.L.; Yang, M.C. Evaluation of Chitosan/PVA Blended Hydrogel Membranes. J. Memb. Sci. 2004, 236, 39–51, doi:https://doi.org/10.1016/j.memsci.2004.02.005.
  2. Guo, Z.; Zhang, D.; Wei, S.; Wang, Z.; Karki, A.B.; Li, Y.; Bernazzani, P.; Young, D.P.; Gomes, J.A.; Cocke, D.L.; et al. Effects of Iron Oxide Nanoparticles on Polyvinyl Alcohol: Interfacial Layer and Bulk Nanocomposites Thin Film. J. Nanoparticle Res. 2010, 12, 2415–2426, doi:10.1007/s11051-009-9802-z.
  3. Hasan, M.W.; Haseeb, M.; Ehsan, M.; Gadahi, J.A.; Naqvi, M.A.-H.; Wang, Q.Q.; Liu, X.; Lakho, S.A.; Yan, R.; Xu, L.; et al. Nanoparticles (PLGA and Chitosan)-Entrapped ADP-Ribosylation Factor 1 of Haemonchus Contortus Enhances the Immune Responses in ICR Mice. Vaccines 2020, 8, 726, doi:10.3390/vaccines8040726.
  4. Wang, Q. Nanoparticles of Chitosan / Poly ( D , L-Lactide-Co- Glycolide ) Enhanced the Immune Responses of Haemonchus Contortus HCA59 Antigen in Model Mice. 2021, 3125–3139.
  5. Haseeb, M.; Huang, J.; Lakho, S.A.; Yang, Z.; Hasan, M.W.; Ehsan, M.; Aleem, M.T.; Memon, M.A.; Ali, H.; Song, X.; et al. Em14-3-3 Delivered by PLGA and Chitosan Nanoparticles Conferred Improved Protection in Chicken against Eimeria Maxima. Parasitol. Res. 2022, 121, 675–689, doi:10.1007/s00436-021-07420-4.

Reviewer 4 Report

Comments and Suggestions for Authors

PLGA encapsulation offers many advantages for antigen delivery because it is biocompatible and biodegradable; can protect antigens from degradation and clearance; allows co-encapsulation of antigens and immune modulators; can be targeted to antigen-presenting cells; and its particulate nature can enhance uptake and cross-presentation by mimicking the size and shape of an invading pathogen. This manuscript describes PLGA encapsulation of ES-15 antigen of H. contortus.

- It is not clear if the antigenic protein used in the study was produced by the authors themselves or if it was provided by a commercial supplier or another laboratory and only encapsulated, if so it should be clarified, otherwise all methods of development from cloning, expression, purification and characterization should be included.

-Improve scientific writing and English used in the manuscript.

-Title should be changed if the main aim of the study is only PLGA encapsulation of antigen rather than development of a new vaccine. Also, most of the data don't show differences between non/encapsulated antigen.

Title for example : PLGA encapsulation of Haemonchus contortus ES-15 antigen improves immune responses in mice

-Abstract needs improvement. Introduction is unclear with insufficient literature review. After brief introduction of the parasites, the ways of its control including theopathic treatment and vaccines should be explained and then challenge of vaccine development and the way of improvement like nano encapsulation should be explained. Previous literatures on vaccine delivery for H. contortus and must have glance on PLGA advantages and literatures on for vaccines delivery. Finally the main goal whether is development or just encapsulation. 

Just focus on vaccine delivery rather than drug delivery!

M & M:

Separate section for regent is not necessary instead can just be mentioned in each relevant procedures section.

As already mentioned, it's not clear whether vaccine is developed in this study or just encapsulated by the authors. If so, it should be clarified, otherwise all methods of development from cloning, expression, purification and characterization should be included.

If the main aim is just to encapsulate an already produced protein, this should be mentioned in all sections from title, abstract, methods, etc... and it is important that in graph and figures in addition to significance of difference with mock injected group, the difference between non and encapsulated antigen must be shown.

2.5. Synthesis of rHcES15 Antigen-Loaded and Blank PLGA NPs 

Change to :  PLGA  encapsulation of rHcES15 Antigen

And 2.6.1. Encapsulation Efficiency (EE), Loading Capacity (LC), and in vitro Cumulative 149

Release (CR) of rHcES-15 Antigen should be moved to this section

2.6. Characterization of PLGA-rHcES15 NPs

2.6.1SEM

2.6.2 DLS

2.6.3 SDS-PAGE

Western blot assay must be included in this section.

Immunization experiment design: It is not clear why pET-32a was considered as a control? While this is a recombinant protein rather than a recombinant DNA vaccine! the study should include the 1-PLGA, 2-antigen, 3-PLGA-antigen and mock injection with PBS, and was better to have a non-immunogenic control protein such as iRFP and/or non-injected control.

Table2 is not necessary, just briefly explain in the text with exact mount of injected doses and relevant methods.

Results:

Fig.1 Figure 1. The surface characteristics, dimension, and zeta potential of antigen-loaded nanoparticles 281 (NPs) were evaluated using sodium dodecyl sulfate–polyacrylamide gel electrophoresis (SDS-PAGE), demonstrating the antigen's integrity post-incorporation into the polymeric matrix.  How using SDS can evaluate zeta potential, surface…!!!!!!!!!!

pET32a is not necessary in the graphs. Once again, the statistical analysis should be checked, as there seems to be some error, for example, in Fig. 3.D, both non and encapsulated antigen show the same value, while the author declared significant differences.

One-way Duncan's ANOVA was performed to compare all group means, and Dunnett's multiple comparison test can be used between each treatment and control mean.

Size measured by DLS and SEM are the same? For this you can image j and measures diameter of the number of NPs in SEM then compare with DLS, showing whether there is significant difference. 

Discussion needs significant improvement; all results should be explained one by one and compared with previous studies and finally the reasons for differences should be discussed.

Table 2 in the Discussion section! Move to Results section.

Comments on the Quality of English Language

Scientific writing and English used in the manuscript need significant improvement.

Author Response

PLGA encapsulation offers many advantages for antigen delivery because it is biocompatible and biodegradable; can protect antigens from degradation and clearance; allows co-encapsulation of antigens and immune modulators; can be targeted to antigen-presenting cells; and its particulate nature can enhance uptake and cross-presentation by mimicking the size and shape of an invading pathogen. This manuscript describes PLGA encapsulation of ES-15 antigen of H. contortus.

Comment 1: It is not clear if the antigenic protein used in the study was produced by the authors themselves or if it was provided by a commercial supplier or another laboratory and only encapsulated, if so it should be clarified, otherwise all methods of development from cloning, expression, purification and characterization should be included.

Author response: Your query is deeply appreciated. Line #145 explicitly states that the authors acquired the purified protein from a colleague after the comprehensive characterization process, which involved cloning, expression, blotting, and purification.

Comment 2: Improve scientific writing and English used in the manuscript.

Author response: Thank you very much for your suggestion. The complete manuscript is revised with more emphasis on scientific English language.

Comment 3: Title should be changed if the main aim of the study is only PLGA encapsulation of antigen rather than development of a new vaccine. Also, most of the data don't show differences between non/encapsulated antigen.

Title for example: PLGA encapsulation of Haemonchus contortus ES-15 antigen improves immune responses in mice.

Author response: The title of this manuscript is revised in the main text as you suggested. We appreciate your valuable recommendation.

Comment 4: Abstract needs improvement. Introduction is unclear with insufficient literature review. After brief introduction of the parasites, the ways of its control including theopathic treatment and vaccines should be explained and then challenge of vaccine development and the way of improvement like nano- encapsulation should be explained. Previous literatures on vaccine delivery for H. contortus and must have glance on PLGA advantages and literatures on for vaccines delivery. Finally, the main goal whether is development or just encapsulation. 

Author response: We extend our gratitude for the valuable comments provided by the reviewer. Notably, extensive revisions have been implemented in the abstract and introduction sections of this manuscript, aligning with the recommendations suggested. Your insights have been pivotal in enhancing the manuscript's quality, and we deeply appreciate your contribution in this regard.

Comment 5: Just focus on vaccine delivery rather than drug delivery!

Author response: It has been revised in the revised version of manuscript. Thank you for suggestion.

M & M:

Comment 6: Separate section for regent is not necessary instead can just be mentioned in each relevant procedures section.

Author response: It has been revised in the revised version of manuscript. Thank you for suggestion.

Comment 7: As already mentioned, it's not clear whether vaccine is developed in this study or just encapsulated by the authors. If so, it should be clarified, otherwise all methods of development from cloning, expression, purification and characterization should be included.

Author response: The retrieval of the purified protein subsequent to its cloning, expression, purification, and characterization from the Laboratory of Molecular Parasitology and Immunology, Nanjing Agricultural University, China PR, has been previously indicated in Line #145 of the main text file. We sincerely appreciate your comment.

Comment 8: If the main aim is just to encapsulate an already produced protein, this should be mentioned in all sections from title, abstract, methods, etc... and it is important that in graph and figures in addition to significance of difference with mock injected group, the difference between non and encapsulated antigen must be shown.

Author response: Thank you very much for your kind suggestion. The figures are changed and the difference between non and encapsulated antigen is evident in revised version of manuscript.

Comment 9: 2.5. Synthesis of rHcES15 Antigen-Loaded and Blank PLGA NPs 

Change to: PLGA encapsulation of rHcES15 Antigen

Author response: It has been revised in the main text file. We highly thankful to you for your valuable suggestion.

Comment 10: And 2.6.1. Encapsulation Efficiency (EE), Loading Capacity (LC), and in vitro Cumulative Release (CR) of rHcES-15 Antigen should be moved to this section. 2.6. Characterization of PLGA-rHcES15 NPs

2.6.1SEM

2.6.2 DLS

2.6.3 SDS-PAGE

Author response: Section 2.5 delineates the process involved in synthesizing rHcES-15+ PLGA NP, whereas the analysis of SEM, DLS, SDS-PAGE, Encapsulation Efficiency (EE), Loading Capacity (LC), and in vitro Cumulative Release (CR) specifically revolves around assessing the characteristics of the antigen-loaded NP. Consequently, recognizing this distinction, the authors have segregated these assessments into a separate section, namely section 2.6. We sincerely appreciate your valued recommendation.

Comment 11: Western blot assay must be included in this section.

Author response: Thank you very much for your kind suggestion. The western blot is included in the revised version of the manuscript.

Comment 12: Immunization experiment design: It is not clear why pET-32a was considered as a control? While this is a recombinant protein rather than a recombinant DNA vaccine! the study should include the 1-PLGA, 2-antigen, 3-PLGA-antigen and mock injection with PBS, and was better to have a non-immunogenic control protein such as iRFP and/or non-injected control.

Author response: We express our sincere gratitude for your suggestion. The rationale behind employing the pET-32a vector protein as a control in this study stems from the cloning of recombinant HcES-15 with the pET-32a vector protein in the prokaryotic (BL21) expression system. Consequently, the incorporation of the pET-32a vector protein was imperative within our experimental design for this investigation.

Comment 13: Table2 is not necessary, just briefly explain in the text with exact mount of injected doses and relevant methods.

Author response: Thank you very much for your recommendation. Table2 is removed from the main text file.

Results:

Comment 14: Fig.1 Figure 1. The surface characteristics, dimension, and zeta potential of antigen-loaded nanoparticles 281 (NPs) were evaluated using sodium dodecyl sulfate–polyacrylamide gel electrophoresis (SDS-PAGE), demonstrating the antigen's integrity post-incorporation into the polymeric matrix.  How using SDS can evaluate zeta potential, surface…!!!!!!!!!!

Author response: Authors extend their gratitude for highlighting the query. The error, which was a result of typing, has been rectified within the main text file.

Comment 15: pET32a is not necessary in the graphs. Once again, the statistical analysis should be checked, as there seems to be some error, for example, in Fig. 3.D, both non and encapsulated antigen show the same value, while the author declared significant differences.

Author response: We greatly value the reviewer's comment. Revisions to the figures have been made within the main text document. The justification for the inclusion of pET-32a has already been elucidated in the preceding text.

Comment 16: One-way Duncan's ANOVA was performed to compare all group means, and Dunnett's multiple comparison test can be used between each treatment and control mean.

Author response: We highly appreciate this comment of reviewer. The figures are revised in the main text file accordingly.

Comment 17: Size measured by DLS and SEM are the same? For this you can image j and measures diameter of the number of NPs in SEM then compare with DLS, showing whether there is significant difference. 

Author response: Thank you for your suggestion. In the upcoming manuscript, the authors will conduct the comparison mentioned above. This manuscript primarily focuses on the immunogenicity of the antigen; therefore, the main emphasis was placed on evaluating the antigen's immunogenic properties rather than on the characteristics of nanoparticles.

Comment 18: Discussion needs significant improvement; all results should be explained one by one and compared with previous studies and finally the reasons for differences should be discussed.

Author response: Thank you for your valuable suggestion. The revised version is changed accordingly.

Comment 19: Table 2 in the Discussion section! Move to Results section.

Author response: Thank you for your valuable suggestion. In comment #13, you suggested removing Table 2 from the manuscript, and here it was advised to move it from the Discussion to the Results section. However, the authors did not remove Table 2; instead, they moved it from the Discussion to the Results section.

Comments on the Quality of English Language

Comment 20: Scientific writing and English used in the manuscript need significant improvement.

Author response: The revised version of the manuscript shows extensive changes that have been made to overcome this shortcoming. We highly appreciate your suggestion.

Reviewer 5 Report

Comments and Suggestions for Authors

The manuscript presents a vaccine development against Haemonchus contortus infection. The objective is of important practical agricultural reason. A nanoparticle vaccine may represent an advanced solution for fighting against Haemonchus contortus infection. The manuscript describes the steps of vaccine development and testing in an outbred mouse model. The chosen methods are adequate and the results are well demonstrated. The specific immune response elicited by the nanoparticle vaccine in the mouse model is encouraging for future practical use. The question is whether a similar immune response that eliminates/prevents infection can be induced in small ruminants?

Author Response

The manuscript presents a vaccine development against Haemonchus contortus infection. The objective is of important practical agricultural reason. A nanoparticle vaccine may represent an advanced solution for fighting against Haemonchus contortus infection. The manuscript describes the steps of vaccine development and testing in an outbred mouse model. The chosen methods are adequate and the results are well demonstrated. The specific immune response elicited by the nanoparticle vaccine in the mouse model is encouraging for future practical use.

Comment 1: The question is whether a similar immune response that eliminates/prevents infection can be induced in small ruminants?

Author response: Thank you very much for your comments.  Another experiment has been done in goat, the similar results were obtained and would be published in the future. 

Round 2

Reviewer 4 Report

Comments and Suggestions for Authors

There is improvement in the manuscript, but still some minor corrections required then can be published.

Since your explain on source and declaration on the main aim of your study, ie evaluation of immunity this recombinant protein rather than compering the encapsulation and your results this title does not exactly resents your work and results and I recommend to keep previous one with minor correction.

PLGA Encapsulated Haemonchus contortus ES-15 antigen improves immune responses in mice.

Table 1 is not necessary, just briefly explain in the text with exact amount of injected doses and relevant methods or can also move to supplement.

There is no reasonable reason to include vector as a control in your study since you didn't evaluate a DNA vaccine where the vector is administered directly with the insert.

Western blotting confirmation added I understand it would be difficult to do new experiments but for your future works you can have both WB of non/ and encapsulated protein showing the stability of your protein after encapsulation. If you have also would be ideal to add to Fig. 1.

Comments on the Quality of English Language

Minor revision required.

Author Response

There is improvement in the manuscript, but still some minor corrections required then can be published.

Comment 1: Since your explain on source and declaration on the main aim of your study, ie evaluation of immunity this recombinant protein rather than compering the encapsulation and your results this title does not exactly resents your work and results and I recommend to keep previous one with minor correction.

PLGA Encapsulated Haemonchuscontortus ES-15 antigen improves immune responses in mice.

Author Response: Thank you very much for your kind suggestion. The title of this manuscript has been changed with the previous one after minor modification.

Comment 2: Table 1 is not necessary, just briefly explain in the text with exact amount of injected doses and relevant methods or can also move to supplement.

Author Response: It is changed in the revised version of manuscript. We are highly thankful to you for your suggestion.

Comment 3: There is no reasonable reason to include vector as a control in your study since you didn't evaluate a DNA vaccine where the vector is administered directly with the insert.

Author Response: We are extremely grateful for your demonstrated concern, which we deeply appreciate. In this investigation, pET-32a serves as a tag protein control (18kDa). It is not DNA when we used in the experiment. As illustrated in Figure 1, the observed band at approximately 33 kDa (ES-15=15kDa+pET-32a=18kDa) signifies the fusion product.  We evaluate this vector protein to discern the diverse implications of pET-32a vector protein, encompassing its potential toxicity, inadvertent interference, and its influence on the immunogenic properties of our target protein (ES-15).

Comment 4: Western blotting confirmation added I understand it would be difficult to do new experiments but for your future works you can have both WB of non/ and encapsulated protein showing the stability of your protein after encapsulation.  If you have also would be ideal to add to Fig. 1.

Author Response:Thank you immensely for your understanding.  Above mentioned figure has been moved to figure 1 in the revised version of manuscript. Moreover, authors are highly thankful to you for your kind advice.

Comments on the Quality of English Language

Minor revision required.

Author Response: It is improved in the revised version of manuscript. Thank you very much for your kind recommendation.